# Exploration of Chitinous Scaffold-Based Interfaces for Glucose Sensing Assemblies

**DOI:** 10.3390/polym11121958

**Published:** 2019-11-28

**Authors:** Dipali R. Bagal-Kestwal, Been-Huang Chiang

**Affiliations:** Institute of Food Science and Technology, National Taiwan University, No.1, Roosevelt Road, Section 4, Taipei 10617, Taiwan

**Keywords:** biopolymer, biosensor, chitin, chitosan, electrochemical sensing, enzyme, glucose, glucose oxidase, interface, nanocomposite, nanomaterial scaffold

## Abstract

The nanomaterial-integrated chitinous polymers have promoted the technological advancements in personal health care apparatus, particularly for enzyme-based devices like the glucometer. Chitin and chitosan, being natural biopolymers, have attracted great attention in the field of biocatalysts engineering. Their remarkable tunable properties have been explored for enhancing enzyme performance and biosensor advancements. Currently, incorporation of nanomaterials in chitin and chitosan-based biosensors are also widely exploited for enzyme stability and interference-free detection. Therefore, in this review, we focus on various innovative multi-faceted strategies used for the fabrication of biological assemblies using chitinous biomaterial interface. We aim to summarize the current development on chitin/chitosan and their nano-architecture scaffolds for interdisciplinary biosensor research, especially for analytes like glucose. This review article will be useful for understanding the overall multifunctional aspects and progress of chitin and chitosan-based polysaccharides in the food, biomedical, pharmaceutical, environmental, and other diverse applications.

## 1. Introduction

Chitin is the second most abundant natural structural polysaccharide after cellulose which is derived from exoskeletons of crustaceans, cell walls of fungi and insects [1,2,3,4,5]. Chitin (CT) and chitosan (CS) are considered as chemical analoges of cellulose, where the hydroxyl groups at the carbon-2 position are replaced by acetamido and amino groups, respectively [6]. Although CT and CS are the collective names for the family of de-N-acetylated chitin with various degrees of deacetylation [3,4,5], chemically, CS and CT are not single entities, but vary in composition, depending on the origin and manufacture process. CT is a long chain polymer, composed of β(1→4) linked 2-acetamido-2-deoxy-β-d-glucose units also known as N-acetyl-d-glucosamine where the N-acetyl-glucosamine units exceed 50% [7]. While CS is aminopolysaccharide polymer which is prepared by deacetylation of CT and is consequently a copolymer of N-acetyl-d-glucosamine and d-glucosamine. Although CS is derived from CT and the N-acetylglucosamine content is less than 50%, CS is the preferred immobilization matrix due to its distinct chemical and biological properties [2,7,8]. Acid soluble CT and its derivatives are often referred as animal cellulose [9] and they are versatile materials that can be used in various fields including environmental [10], foods-dietary supplements [11], pharmaceuticals-cosmetics [12], textiles, water treatment and coatings applications [9,10,11,12,13,14,15,16,17,18] (Figure 1). The most prominent properties of both CT and CS include biodegradability, bioactivity, biocompatibility, film coating ability, high miscibility, eco-friendly, nontoxicity and non-allergic [19,20,21,22]. These properties are highly desirable for enzyme immobilization, electrode or transducer modification, development and application of biosensor/sensing system for many applications [23].

In this review, we focus on various innovative strategies used for the fabrication of biological assemblies using chitinous-biomaterial interface for glucose sensors and their applications in food, biomedical and other areas. Different roles of CS/CT such as a protective polymer, immobilization matrix, interface modulators and transducer-amplifiers for biosensing devices will be discussed. The literature studies are categorized into: Blend or composites of CT/CS, CT/CS nanocomposites and the techniques for designing the interface based on enzyme functionalization and their influence on sensor communication. The application for glucose detection using glucose oxidase (GOx) in the biomedical field, food industry, environment, and other sections will be discussed (Figure 2). This review will be useful for understanding the overall progress and role of these polysaccharides in the interdisciplinary biosensor research especially for analytes like glucose.

## 2. Composites of Chitin and Chitosan

The polymer composite consisting of two or more chemically and physically different materials is considered as a wonder material because it can yield a distinct new interface with structural and functional superiority [24,25]. Although the composites are combinations of several different materials, the individual constituents still retain their properties/identities. The primary matrix/phase is the continuing matrix which holds the embedded secondary dispersed phase/reinforcement phase [25]. The CT/CS composites are also known as green composites with ecofriendly, fully degradable and sustainable properties. Solution mixing, melt mixing, in situ polymerization, dry powder mixing, and aqueous mixing techniques are some of the common but important methodologies to prepare CT/CS composites [1,2,25,26,27]. These composites can be prepared judiciously to get anticipated archetypes with desired physicomechanical, biochemical, morphological, bio-durable and biomimetic properties [26,27]. Composites of conducting polymers and CS have also shown excellent properties as conducting and biocompatible material for various bio-electronic applications. The CS-composites with conducting polymers were mostly prepared by in situ electropolymerization, and characterized by electrochemical measurements, such as FTIR, SEM and AFM analysis [28]. Koev et al. have documented the common methodologies for fabrication, modification, and characterization of CT/CS-composite films-based micro-scale devices and their applications [29].

## 3. Nanocomposites of Chitin and Chitosan

Nanocomposites are composed of reinforced nano-sized materials in a dispersed polymer matrix, typically an organic matrix comprising inorganic nanoparticles/structures within [30]. Nanomaterials are one of the most essential components for enzymatic sensor devices due to their large surface areas, super catalytic conductivity and signal amplification ability [31,32,33,34]. Although agglomeration/aggregation due to high surface energy and poor binding to the substrate of these unique and diminutive nanostructures sometimes limit their use for analyte detection in sensing systems, these limitations can be easily overcome by using suitable matrix and nanomaterials as reinforced materials. CT and CS are used as matrix phase when blend or blend with other polymeric materials [26,27,33,35]. Most of the properties of such composites directly depend upon the reinforcing phase/material’s properties, which need to meet the practical purposes. These nanocomposites with special intrinsic properties, mainly regulated by nanostructures, can be produced by bottom-up or top-down methodologies including vapor phase deposition (VPD) and in situ synthesis [19,26,27,30,36,37,38,39]. Nanocomposites of CT/CS is a multiphase material where enzymes and nanomaterials are incorporated into CT/CS matrix using various methods. We will discuss the impact of nanomaterials integrated chitinous polymer for glucose biosensors in this review article.

## 4. Method of Preparation of Chitinous Nano-Structures

Nano derivatives of crustacean materials can be prepared by several methods, depending upon the source of chitin. For example, chitin nanowhiskers are commonly produced by hydrochloric acid hydrolysis [40], precipitation, ultra-sonication [41], mechanical treatment [42], spray drying [9,43], ionotropic gelation [44,45,46,47,48], emulsion-droplet coalescence and reverse micellar method [49,50]. While TEMPO method [51,52], electrospinning and a simple grinding treatment [53] were also used to obtain chitin nanowhiskers [41]. Routinely, CS nanoparticles are prepared by degradation to low-molecular-weight chitosan using hydrogen peroxide (H_2_O_2_), then cross-linking is followed by the treatment of tripolyphosphate (TPP) [53,54,55,56]. The nanochitosan (NCS) was also produced based on ionotropic gelation between low molecular weight CS and TPP under microwave-irradiation [55]. Various methods for preparing different nano-forms of CT and CS nanoparticles, nanowhiskers, nanofibers, etc. have been well documented by Divya and Jisha, 2018 [10].

## 5. Chitinous Scaffold for Immobilization

Chitinous nanocomposites have great potential for biomedical, pharmaceutical and other versatile applications. The desired properties of these engineered materials such as non-toxicity, biocompatibility, biodegradability, and low-allergenicity led to an increased interest in the use of implantable biosensors for continuous monitoring of physiological biomarkers. Due to superior physical properties including film-forming tendency, high surface area, porosity, tensile strength and conductivity, they can be easily molded into functional interfaces [57]. Enzyme immobilization is confinement of enzyme to the matrix or support which allows the enzyme to stay active for a longer time with possibilities of multiple uses without compromising its biological activity. Amenable reactive amino and hydroxyl groups, cationic polyelectrolyte nature and acid solubility of CS, satisfy most of the demands of enzyme immobilization and biosensor fabrication for many commercial applications. The derivatives or composites of these chitinous supports are also prepared as per the demand of biomedical or food industrial applications. Therefore, chitinous materials in various configurations such as powders, flakes, beads, membranes/films, capsules, fibers, sponges, lyposomes and sol-gels have been used as the support for enzyme immobilization [7,8,58]. Here we will discuss the CS and CT-derived composites and nanocomposites for enzyme immobilization and fabrication of biosensing devices for practical applications.

### 5.1. Chitin as an Enzyme Matrix

A biopolymer like CT offers remarkable functional and divergent biological properties based on electrostatic interaction with other materials. The protonation of the acetylamide group of CT helps enzyme immobilization due to its negative charge. An electrochemical sensor for glucose determination in a sport drink was developed using a modified carbon paste electrode (CPE). A thin film of immobilized GOx-CT-platinum (Pt) powder was developed at the CPE surface for detecting H_2_O_2_ produced from glucose [59]. The same research group also presented another glucose sensor using a modified platinum electrode (PtE) with a chitin-glucose oxidase (CT-GOx) film. The constructed electrochemical sensor can detect glucose content ranged from 5 × 10^−7^ to 3 × 10^−5^ mol·dm^−3^ [60]. The effective adsorptive equilibrium between the electrode surface and electrolyte showed constant current response without enzyme leakage. The strong adsorption of the enzyme on CT is due to electrostatic interactions of GOx with CT within the film.

### 5.2. Chitosan as an Enzyme Scaffold

The CS also has many promising biocompatibility characteristics, including excellent adhesion property, high mechanical strength, and tunable functional groups for chemical modifications [3,61]. The environmental friendly CS is an ideal low-cost matrix. It meets the current market demands because of its easy production, less immunogenicity, nontoxicity, high biodegradability and desirable stability [61,62]. These superior properties have impelled extensive applications of CS as a support/matrix for enzyme immobilization. The GOx conjugated with CS can improve its resistance to chemical degradation, reduce leakage, and interference with metal ions. Reactive amino and hydroxyl groups of CS provide good coupling efficiency with biological entities, including enzymes. The stable interactions with CS help to prevent the enzyme from leakage when altering the diffusion rate [62,63,64,65,66]. CS plays a decisive role in the maintenance of the immobilized biocatalyst and has gained tremendous interest in drug delivery technology without any cost-constraints. CS can also improve the thermal and storage stability of the immobilized enzyme by providing biocompatible microenvironment [66,67,68]. CS-GOx in the form of microspheres showed remarkable storage stability with high encapsulation efficiency. The redox enzyme was encapsulated in calcium alginate-CS microspheres using an emulsification-internal gelation-GOx adsorption-CS coating method [69]. In another study, composites of carrageenan (*κ*-, *ι*-, *λ*-) and CS were prepared for micro-encapsulation to protect GOx under acidic conditions [70].

### 5.3. Chitosan Cryogel

Cryogel is a form of a gel formed by freeze-drying a sol. The porous structure of cryogel provides a large surface area for biomolecule immobilization, which also has a significant impact on enzyme activity and reusability [71,72,73]. The parameters including CS polymer ratio, amount of cross-linker, the temperature of the cryogelation process, and stirring conditions would directly influence the resulting gel′s chemistry, pore morphology, microsphere size, swelling behavior, and degradation rate [74,75,76,77,78,79,80]. Recently, injectable chitosan cryogel microsphere scaffolds in the form of microspheres were synthesized using a water in oil emulsification method. The cryogel microspheres that were further cross-linked using glutaraldehyde (GA) had an average pore and particle size of 5.50 ± 0.63 and 220.11 ± 25.58 μm. It also showed that the as-prepared CS-cryogel is highly suitable for different noninvasive tissue engineering applications [75]. In another study, a CS cryogel-based sensor was fabricated to improve the biosensor performances, mainly their sensitivity and stability during glucose detection [72]. The porous CS cryogel beads with a large surface area were proven to be an excellent matrix for enzyme immobilization by cross-linking with sodium tripolyphosphate at a subzero temperature [74]. The permeable CS cryogel also allowed the test solution to flow through easily and the analytes diffused freely over the surface and into the gel pores to interact with the enzyme, ultimately reduced the reaction time. Hedström et al. reported a novel monolithic macroporous material that was developed by cross-linking hen egg albumin (HEA) and CS with GA at subzero temperatures [80]. This matrix was used for immobilization of various enzymes, including GOx, HRP, savinase, and esperase for flow injection analysis (FIA). The low millimolar range detection of glucose was achieved after GOx was covalently coupled with the CS-HEA matrix during FIA, proving enzyme retention capability of HEA-CS-GOx for preparative applications [79].

### 5.4. GOx-Chitosan Electrochemistry

Commercial glucose sensors for daily blood sugar detection is a great breakthrough. However, these point-of-care devices do suffer from high cost due to one-time use, low sensitivity, and interference problems. Interference from other electroactive species in physiological sample is the main challenge for these sensors. Therefore, tremendous efforts have been dedicated to establishing an interference-free/resistant interface for these biosensing devices. The bioelectronics interface may advance the transducer’s efficiency though it’s the inherent selectivity or specificity to amplify the analytical signal. Therefore, an important thing is to achieve the best biochemistry between the interface matrix and enzyme, which also comply with the morphological demands of these fragile biological entities [81]. The ideal interface also needs to fulfill the demands like fast communication, high signal amplification capabilities, and reusability issues. Apart from the communication challenge, these bio-interfaces should have the bio-mimicking ability or background to facilitate the canalization of the biological process at the device for active molecules such as enzymes.

Chitosan is known to be a perfect material for GOx immobilization as it provides a protective shield to the fragile enzyme molecules. The amino group of CS makes it more suitable for enzyme immobilization as compared to the other polymers. Chitosan accommodates the three dimensional (3-D) conformation of GOx by providing comfortable hydrophilic vicinity and structural stability. CS also provides pseudo cellular conditions via the cross-linked network of polymeric chains which directly or indirectly maintains the enzyme-friendly microenvironment. For example, electropolymers such as poly(N-methypyrrol) (PNMP) were used to synthesize film at the PtE surface to fabricate a reusable glucose biosensor. CS-GA-GOx was dip-coated to prepare a second film on the PNMP film on the electrode. The leakage of the enzyme was prevented by cross-linking immobilization which was confirmed by 20 cycles of reuses with retention of 91.3% of initial current response [82]. Sol–gel approach at ambient conditions was used for the preparation of organic-inorganic hybrid gel by entrapping GOx in CS with tetramethoxy silicane (TMOS). This study confirmed that the highly porous micro-structured GOx-TMOS composite not only provided a high immobilization yield (97%) but also prolonged bio-stability (15-day) at 30 °C [83]. The composite sol-gel film of methyltrimethoxysilane (MTMOS)-CS-GOx was coated on the ferrocene-modified glassy carbon electrode (GCE) for electrochemical sensor fabrication. This modulated composite material also provided good biocompatibility and good GOx stability by alleviating the adverse microenvironment around the enzyme [84]. Another example is the sol-gel composite of CS and silica (Si). The CS/Si composite film showed improved features including physical rigidity, tunable porosity, and chemical inertness because of the silica, while the controlled shrinking-swelling behavior, pH stability, and thermal resistance could be attributed to the CS. The biosensor fabricated under optimal conditions had a fast response time, superior sensitivity and long-term stability of over 60 days with good substrate selectivity which again owing to the CS′s unique properties [85]. The submicron particles of chitosan from gladius of *Todarodes pacificus* were produced by ball milling technique, they showed high affinity and good biocompatibility for GOx [86]. In this case, CS was derived from β-chitin using deacetylation procedure which was followed by ball mill pulverization to form an ultra-fine white powder. When used for enzyme immobilization, the porous structure of ZnO/Pt nanocomposites could promote enzyme binding and provide higher conductive surface and more active sites for the GOx molecules. Another CS-composite was prepared by incorporating the carbon nanotubes (CNTs) with gum Arabics gA) to increase the membrane conductivity and GOx binding capacity. The CS membrane with gA had two-fold enzyme loading capacity as compared with the membrane without gA. The properties such as enzyme loading capacity, pH/thermal/storage stability and reusability of this CS matrix were found superior to the other matrices [87]. In another attempt, a protective film containing the permeability-controlling agent, Acetyl Yellow 9 (AY9), using glutaric dialdehyde as a molecular tether for CS matrix was prepared in single-coatings. These novel films coated on PtE showed potential application in noninvasive and portable device fabrication with ultralow and interference-free glucose sensing [81]. Gao et al. fabricated a wireless magnetoelastic biosensor for detecting glucose in urine using co-immobilized GOx and catalase in CS hybrid. The detection principle of this sensor was based on GOx-catalyzed hydrolyzation of glucose into gluconic acid, resulting in shrinking and corresponding mass decrease in this pH-responsive polymer, which subsequently increases the resonance frequency [88]. CS also played a role as the supporting matrix for co-immobilization of various enzymes. Multi-porous nanofibers (MPNF) of SnO_2_ with high surface area and good electrical conductivity were synthesized by electrospinning and further polymerized with polyaniline (PANI). These polymerized nanofibers were optimized for GOx and HRP conjugation with CS [89]. This MPNFs-based novel sensor showed potential for glucose detection in blood and urine, which is indispensable for the diagnosis of diabetes [89]. In another study, HRP was coupled with GOx for preparation of a fluorometric FIA system for glucose determination in beverages, Japanese sake, and liquors [58]. Liposomes with entrapped GOx were prepared by covalent immobilization using GA-activated CS gel beads for a practical bioreactor application with high stability and reusability features [90]. Thus, various forms of CS matrix showed significant hospitality to host the 3-D structure of biomolecules that would usually be ravaged by harsh chemicals during immobilization. Chitosan/chitin glucose sensing systems based on suitable immobilization methodologies to obtain stable GOx are presented in Table 1.

## 6. Innovative Strategies for Sensor Interface

Enzymes are conjugated with nanomaterials and chitinous matrix to fabricate biosensors. The retention of the morphological features and biological activity are the key for high sensitivity, wide linear range, low LOD, and reproducibility of the devices. Skilled modification and linking of such biomaterial-based interfaces increase electric communication and speedup the reaction rate which ultimately enhance the sensor’s performance. As an example, the following section will discuss various innovative approaches and methodologies for the preparation of probes for glucose sensing.

The first-generation glucose biosensors were based on the use of natural oxygen substrate and the detection of the H_2_O_2_ produced. However, these sensors are highly susceptible for the interference caused by the endogenous electroactive species and fluctuations in the oxygen tension [87,104,105]. The redox mediators were introduced in the sensing assembly in the development of the second-generation sensors. Mediators like ferrocene (Fc), ferricyanide (FCN), quinines (Q), tetrathialfulvalene (TTF), tetracyanoquinodimethane (TCNQ), thionine, methylene blue (MB), and methyl viologen (MV) were used to enhance sensor performance [104]. In the third-generation sensors, immobilized enzyme(s), mediators, nanomaterials and polymers are in direct contact with probe. The design of these wired-interfaces are for direct electrical contact, which helps electron migration between the enzyme’s active site and the working electrode surface in the sensing assembly to generate amplified and rapid response [104,105].

In most of the electrochemical biosensor, GOx-flavin adenine dinucleotide (FAD) redox center catalyzes the electron transfer from glucose to gluconlactone. The communication between GOx-FAD and electrode via direct electron transfer (DET) is a big challenge, especially for the third generation i.e., label-free glucose biosensors. It is well understood that an electron transfer of GOx is controlled mainly by reorganization energies; potential differences and orientations of involved redox active sites and distances between redox-active sites and mediator [84,94]. The two bound redox active FAD cofactors of GOx are deeply buried inside the protein shell which acts as insulating shield. These prosthetic shells act as barriers for DET with bare electrode surface [41,44,84,94,96]. Therefore, nanomaterials are introduced to improve the electro-enzymatic process by enhancing the adsorption of the immobilized enzymes in the vicinity of the transducer probe. These nanostructures also helped to retain the biological catalytic activity of GOx in selected scaffold. The association or interaction of nanomaterials with enzyme can be controlled using various physical properties of the nanostructures to promote DET behaviors of matrix entrapped enzyme.

The surface energy, shape and size of the nanomaterials directly and dramatically improve their conjugation with enzymes, and interface performance and pattern. As shown in Figure 3, inclusion of nanomaterials to enhance the analytical performance of many biosensor designs also lead to a high sensitivity and selectivity toward analytes with relatively low interference. Recent research proved that when nanomaterials were used in combination with enzyme during immobilization and subsequently sensor fabrication provided controlled, fast and enhanced detection sensitivity. Studies have also shown that the chitinous biopolymer in sensor could stabilize nanostructures and facilitate electron transfer to the transducer with high surface energy, it also offers maximum enzyme loading due to increased compatibility. The structure geometry formed at the electrode is a key step for sensor fabrication, which is highly sensitive to the enzyme-nanomaterial deposition conditions, matrix nature, biofunctionality, and immobilization protocol. Therefore, in the following sections we will also discuss the methods to create electrically active yet biologically and practically stable probe-interface assemblies with remarkably improved sensor performance (Figure 4). 

### 6.1. Electrode Material/Refilling Matrix

Self-assembled monolayers (SAMs) or layer-by-layer (LBL) structures often hinder the electron and substrate-product transportation due to compact architecture. Being biodegradable, non-toxic. and highly bio-compatible, CS composites are proved to be the best biorecognition transducer. Some studies investigated the application of chitosan nanocomposites as an electrode matrix/material for glucose sensing where minimum interference at low potential is desired [83]. The increase in surface area, enzyme loading, and other unique and multifunctional properties of the working electrode substrate were notably improved by integrating the chitosan [83,106,107,108]. The electrode-surface controlled reactions not only face electron-traffic problems but also lead to lowering bio-catalytic activity of the immobilized enzymes. Surface-coated electrodes also suffer from leakage or fouling due to the overloading of enzymes/nanomaterials [109]. This serious and practical problem can be resolved by co-modification of electrode matrix with chitosan/nanocomposites of chitosan [26,27,109,110,111]. Modification of electrode with co-immobilized nanomaterials and enzymes with polymer matrix is another experimental strategy to overcome the limitations of compact/layered structure. Chitinous matrix also acts as a binder and 3-D framework when used as electrode packing material. In a study, the black binding string composed of polyester spun coated with PB modified graphite (PB-G) ink was used to fabricate electrode [97]. The GOx and CS were coated on the string electrode (StE) by simple dipping process. The modified StE has properties such as stable reproducibility, reusability, good sensitivity and selectivity with fastest response towards glucose. CS with excellent film-forming ability and high permeability toward solvents showed good adhesion and biocompatibility. It also accelerated the catalytic H_2_O_2_ reduction mediated by PB-G at string surface.

A 3-D porous film of CS and single-walled carbon nanotubes (SWCNTs) were utilized to construct a thin film-electrode after entrapping and cross-linking with GOx. The enzyme immobilized on this CS-SWCNTs film had higher enzyme loading and enzyme activity as compared to the non-porous planar films [112]. Such electrode-assembly has high surface area and interconnected porous structure that could enhance reaction efficiency. The microporous electrode matrix also minimizes substrate diffusion effects which limit the use of multilayer films of enzymes assembled on planar substrates due to thickness, and the electrode efficiency was enhanced by one or more order than that of the corresponding surface modified electrodes. Reagentless glucose sensor based on the excellent electron transfer acceleration rate of CNTs, Fc, and GOx on the electrode matrix was reported by Zhou et al., 2017 [113]. The high loading of the enzyme within the nanocomposite was achieved by covalent link between the positively charged CS particles and negatively charged GOx. An effective biocompatible environment to facilitate electronic communication with improved catalytic nature was possible due to the porous zinc oxide/platinum nanoparticle (ZnO/Pt)NPs-based electrode. Submicron particles from gladius of *Todarodes pacificus* (GCSPs) with porous structure effectively promoted the immobilization of enzyme by providing comparatively larger conductive surface and more active sites for the GOx molecules [86]. Besides, the CT-based paste was made by mixing graphite powder, GOx, Pt powder, and Nujol, and this paste filled the cavity of the carbon electrode [59]. In this study, chitin not only held the biological activity of GOx but also provided longer lifetimes of the fabricated electrode. The electrodes with enzyme-CT polymer-mediator also restrain the leakage of GOx.

Graphene (GR) is a two-dimensional monolayer of carbon atoms bonding with sp2 hybrid orbitals [114]. Qian and Lu (2014), reported that ice-induced assembly of 3-dimentional porous GR-CS composites from freeze-drying as a matrix for enzyme immobilization. This GR-CS-GOx composite with porous and layered structure showed properties like high mass transfer speed, effective electroactive surface, high conductivity and loading capacity when used as sensor [114]. A glucose sensor-based GR-CS-GOx was reported by Kang et al., 2009. The excellence of the sensor was attributed to large surface-to-volume ratio and high conductivity of GR and good biocompatibility of CS. According to the authors, physically absorbed GOx on GCE surface showed superior dispersion stability of GR and excellent catalytic efficiency. The chemical functional groups of hydrophilic graphene (–C–OH, –COOH) well interacted with CS matrix to form stable hybrid structure. The electron-transfer-rate constant (*k*_s_) of the GOx in CS-GR nanocomposite suggested that the modified electrode provides direct electron transfer between the redox center of the enzyme and the electrode surface due to CS-nanostructured confinement [115].

A single nanofiber electrode made up of one-dimensional (1D) mesoporous ZnO/CS inorganic–organic hybrid nanostructure with high enzyme loading and stability features was reported [116] (Figure 5).

This device had mesoporous nanostructure with protuberances which favored enzymes loading via electrostatic adsorption and enhanced electrical communication efficiency. This free probe type biosensor prototype is highly suitable for micro-targets detection in microcell and enzymatic studies. It also has the potential to be inserted into single cell or other microorganism for biological studies in the future.

### 6.2. Bare Electrode Modification

Most of the electrochemical biosensor are based on the enzyme-catalyzed reactions and electron transfer between enzyme(s) and electroactive species with electrodes/transducers. However, one of the big challenges is to DET at the surface of bare working electrode (for example: GCE disc electrode). The insulation-shelled redox center of the bare electrode surface directly hamper the performance of the sensor [117]. Therefore, there are constant efforts to resolve this problem by modification of working electrode with coating, or casting of electrode surface by matrix with or without redox enzymes/matrices, nanomaterials, etc. either in the form of sol-gel coating or thin film attachment. Incorporation of electroactive materials with electrode material has also been investigated for controlled surface interactions and fast analytical performance [26,27,118,119,120,121].

#### 6.2.1. Electrode Surface Coating

Chitosan possesses excellent adhesive film forming property, it has been demonstrated to be the most suitable matrix for inclusion of single or multiple nanostructures via coating techniques. CS sol-gel to form film or membrane, are the first choice to construct an enzyme-electrodes. CS-hydrogel with or without other constituents showed the pH-dependent volume phase transition which was highly useful and effective for enzyme-nanomaterial integration in film layout. This strategy is also convenient for holding its natural properties to construct GOx and other analytical important sensors. Herein, we have discussed some of the examples where different configuration methodologies involving CT/CS films for probe modification to prepare enzyme-electrode interface and their application for either reusable or disposable sensors.

##### Sol-Gel Casting

The 3-D network of CS can be formed by addition of sequester biomolecules/nanomaterials and this thick, uniform sol-gel can be drop cast on the tip/surface of the electrode to form film structure after drying. The porous and adhesive feature of the sol-gel CS is an effective entrapment/encapsulation matrix for the immobilization of biological element such as GOx. The electrode could also be modified by blank CS film formed at the top of the working electrode first, and then enzyme was linked via chemical treatment. Nafion (Nf), butyl carbitol acetate, dimethylformamide, cellulose, cellulose acetate, polyvinyl alcohol, etc. have been used as CS-film binder at electrode surface [31]. Strong physical adsorption and electrostatic interactions between sol-gel matrix and enzyme has been achieved by chemical functionalization using GA or formaldehyde (FA). Simple drop cast [98,114,122], dip-coating [97], spray drying [122], spin coating/casting [123] and shear spreading technique [124] are some of the common coating methods used for manual probe modification. For example, microwave-assisted synthesis of nanocomposite consisting of reduced graphene oxide (rGO), zinc oxide (ZnO) and silver nanoparticles in CS matrix and its reduction for enzyme immobilization was reported. This biosensor was highly selective, well reproducible and stable with the detection limit is 10.6 µM and linearity range of 0.1 to 12.0 mM for glucose [125]. Figure 6 depicts the general methodology used for the construction of an enzyme sensor by modification of electrode. Figure 6a shows the modified electrode prepared by spin casting methodology using CS-nanocomposite core-shell while Figure 6b shows highly stable self-assembled layer by layer formation of N-doped enzyme matrix. The encapsulated bio-nanohybrid film formation at GCE surface can be seen in Figure 6c. Effective immobilization of GOx on CS-submicron particles for amperometric glucose biosensor is illustrated in Figure 6d. The disc electrode such as glassy carbon (GC)/platinum (Pt)/graphene (GR) were modified using CS for glucose sensor. The bare electrode surface was cleaned by polished with alumina first, followed by ultra-sonication in deionized water, and finally subjected to air-drying. Either pure CS or GA-activated CS solution was drop-casted on the electrode surface [101]. After natural drying, enzyme was added for cross-linking to form CS-enzyme membrane using GA/FA. Monolayer coating of CS-graphene oxide (GRO) composite with protein affinity also effectively prevent the leakage of the mediator. High enzyme stability was achieved when two-dimensional plenary sheet with open structure of graphene oxide nanosheets (GRONSs) utilized for supporting enzyme(s) [126]. Another nanocomposite-based electrochemical potential sensor for clinical utility and home care was reported with sensitive detection limit of 0.6 μM [127]. The sensor assembly was constructed using GOx/Pt/functional graphene sheets (FGRS) in CS for a rapid monitoring of glucose. The biosensor also showed good reproducibility, long-term stability and negligible interfering signals from AA and UA with the electrocatalytic synergy of FGS and Pt nanoparticles to H_2_O_2_. New CS-nanocomposite matrix comprising polypyrrole nanotubes decorated with AuNPs was prepared by simple physical entrapment method [124]. The electrochemical analysis revealed that the highly conducting Au-NPs entrapped in 3-D matrix of CS-PPy-NTs were responsible for a surface confined enzymatic process. These metallic nanoparticles had a major effect on the exchange of electron between GOx redox site and ITOE surface. The bioelectrode with quasi reversible behavior also depicted good linearity for glucose [124]. The physisorption of GOx within hetero-structured film of CS-ZnO-PtNPs was partially based on electrostatic interaction between positively charged chitosan biopolymer and negatively charged GOx [128].

The synergistic action of PtNPs, MWNTs and sol-gel of CS-SiO_2_ was proved for amperometric glucose detection. The excellent stability for 50 days with 90% enzyme activity could be attributed to two features. First, as the enzyme was physically entrapped in the CS-SiO_2_ gel, large quantities of hydroxyl and amino groups help to maintain the enzyme activity by forming strong interaction between GOx and hydrogel. Secondly, the natural composite provided a good microenvironment with the bottleneck effect of the silica sol-gel to prevent the enzyme from leaking [129]. Another drop-coated reagentless nanosheets-based platform for electrochemical biosensor was successfully constructed by entrapping GOx in a Fc-branched organically modified silica material (ormosil)-CS-GRONSc composite by Peng et al., 2016 [130].

Gold nanoparticles (AuNPs) decorated CS-GR nanocomposites film was prepared and characterized by Shan et al., 2010 [131]. The gold electrode was modified using dual nanomaterials AuNPs-GR by direct absorption into CS matrix for glucose biosensing. The sensor showed lower detection limit of glucose of 180 μM with good reproducibility when employed for real blood samples. Another amperometric GOx-GCE sensor composed with CS-AuNPs and gold-Prussian blue nanoparticles (Au-PB) NPs coating proved that electrocatalytically prepared layer not only provides satisfactory sensor′s operational stability but also gives wide calibration for glucose detection with fast response. The sensor without CS protection showed the PB desorption from the electrode within 10 min. The CS nanocomposite matrix provides better surface protection for the immobilized enzyme when continuous functioning for more than 2 h. [132]. Furthermore, a homogenous CS with Fc and AuNPs-GOx biocomposite film was deposited by simple and controllable electro-catalytic methods to fabricate glucose sensor. This enzyme-nanocomposite film retained biological activity of the immobilized enzyme due to a biocompatible microenvironment around the biomolecules [133]. Graphite rod (Gr) electrode was modified using CS-stabilized gold-coated iron oxide (CS-Fe_3_O_4_-Au) magnetic nanoparticles and GOx via cross-linking for blood glucose level estimation [134]. The CS-Fe_3_O_4_-Au nanoparticles facilitated the oxidation of H_2_O_2_ at the electrode and exhibited a good selectivity and fast amperometric response. This third generation of sensor technology, the GOx was directly immobilized onto the Gr electrode to improve shelf-life and reusability for practical application in clinical analysis. A potentiometric glucose sensor based on the CS-GOx-Fe_3_O_4_NPs composite modified gold coated glass electrode was reported by Khun et al., 2012 [135]. This work provided an alternative way for the fabrication of glucose sensor without the use of Nafion or cross linker molecules. CS-stabilized enzyme was directly absorbed by iron ferrite nanoparticles due to its large pore size and volume. These particles also provided desirable microenvironment which remains useful for the display of the active center and thereby increases the catalytic activity of enzyme. Fabrication of ZnO-CS-graft-poly(vinyl alcohol) (gPVA) core-shell nanocomposite-based potentiometric biosensor was prepared by a simple two-step spin casting technique. First, a colorless ZnO core-shell CS-gPVA nanocomposite solution was spin-casted on ITO glass plate at the speed of 2000 rpm to make core-shell nanocomposite film. After air drying, thin enzyme layer was constructed using spin-casting equipment at a speed of 500 rpm. The resulting GOx immobilized core-shell nanocomposite electrode was tested for sensor characterization and glucose detection on blood serum and urine samples collected from a healthy person [123]. However, the sensor was suffered from ~10–15% detection error mostly due to the interferences of the electroactive species from real sample. Another core-shell polymeric–metal oxide nanocomposites was prepared by spray-drying technique. The manganese dioxide-core–shell hyperbranched CS provided rapid, efficient and direct electron transfer when used for screen printed electrode surface modification by drop-casting [122].

Dip coating/dipping, spin coating, and blade coating methods are also used for creating various low-cost configurations at electrode surface. Self-assembly of PANI-grafted CS/GOx nanolayered films prepared by LBL-dip coat technique for electrochemical biosensor applications were reported by Xu et al., 2006 [136]. These hydrophilic and biocompatible films showed a rapid response and high sensitivity. A higher response current to glucose can be achieved with increasing in number of layers. The different layered patterns/configurations and other preparation methods for sensor improvement are summarized in Table 2. Some sensors with notable features from previous research are also discussed.

##### Electrochemical Deposition

Electrodeposition is another conventional surface modification method to improve the surface characteristics and functional properties of a wide variety of materials including biomolecules [137,138,139,140,141,142,143,144,145,146,147]. The surface of a conductive composite material can be immersed in a sol-gel solution containing a metal salt, and by the action of an electric current the coated films with controllable thickness could be obtained [148]. This technique also allows the co-deposition of enzyme and nanomaterials onto transducers for the fabrication of biosensors. Reactive, patterned layer of CS-GOx-AuNPs was electrodeposited on the gold disk electrode surface (cathode) for biosensor construction. The biosensor covered with NF was successfully applied to serum sample analysis [149]. Chitosan’s electrodeposition is a simple and versatile method to form thin-film assembly for the fabrication of optical/electrochemical or mass-based biosensors [150]. The multi-parameter electrochemical quartz crystal microbalance (EQCM) based on crystal electroacoustic impedance analysis was used to dynamically monitor the deposition processes of MWCNTs-CS-GOx nanocomposite films on Au substrate. Zhu et al., 2007 reported the electroreduction protocol where oxidants (p-benzoquinone) or H_2_O_2_) were added for wider applications in development of pH-sensitive composite films for sensing. Authors have achieved simple, fast, uniform, and controllable co-deposition of CS-hydrogel, GOx, MWCNTs for constructing biosensors [151]. In another attempt, a stainless steel needle electrode (SSN electrode) was modified using co-electrodeposition of Pt–Pb nanoparticles and then CS-GOx biocomposite with benzoquinone (BQ). This bio-electrodes showed accelerated electron transfer rate and larger effective surface area of the working electrode. This electrode also has a low cytotoxicity and may have possible application for in vivo uses [152].

Gold disk electrode (AuDE) was modified by CS film integrated with manganese dioxide (MnO_2_NPs) and entrapped GOx, this electrode showed high resistance to the ascorbic acid (AA) during electrochemical glucose estimation. CS-stabilized MnO_2_NPs directly suppressed the interfering signals of ascorbates by converting AA to an inactive product before it reached the electrode surface. Thus, the proposed CS-GOx/ MnO_4_NPs/AuDE sensor is another possibility for the fabrication of interference-free glucose biosensors [158]. In another study, the synergistic influence of AuNPs, MWCNTs, CS and ionic liquid (IL) on direct electron transfer between GOx and electrode was achieved within six seconds. Chronoamperometry, CV and EIS were used during GOx sensor analysis which precludes interferences from UA and AA [153,154,155,156,157,158,159,160]. Nano-gold electrode was prepared by CS-IL-GOx electrodeposition for glucose detection in serum samples. The developed sensor showed 20-fold sensitivity, wider detection range and better anti-interference ability as compared with the plain nano-gold electrode. Electrochemical properties of IL such as non-volatility, non-flammability, high ionic conductivity, thermal stability and wide electrochemical window might have contributed to the excellence of nano-gold biosensor [154]. However, voids and bubbles of the CS-enzyme embedded films are the most significant disadvantage. Such deformation arises during the electrodeposition process due to hydrogen gas formation from the reduction of water/protons [161,162,163]. The bubble/void formation can be reduced by using proton consumers such as BQ/ chloramphenicol (CAM).

#### 6.2.2. Various Geometrical CS-Based Interfaces

##### Self-Assembled Monolayers

As mentioned above, drop, dip, spin coating and spread/blade spreading methodologies are used for preparing aligned self-assembled monolayers (SAMs) to smooth the ruff electrode surface using CS matrix. In some cases, entrapment/encapsulation/electrochemical deposition/electropolymerization have been performed using hybrid solution of enzyme-chitosan and nanomaterial to construct mono or multi-layered film structure [44,131,136,148,155,156,157,158,159]. CS-based SAMs provides great benefit to enzyme in a number of ways as they can be designed in various geometrical shapes depending upon the interface (Figure 6). These SAMs prevent non-specific adsorption of species (interferons), shield enzyme and improve the analyte sensitivity. The nanoscale individual GONSs were acted as “molecular wires” to connect the active sites of GOx and electron mediator Fc with the electrode via self-assembled membrane/film structure to increase the electron transfer rate significantly [126,130]. Furthermore, these SAMs also improved film morphology, thickness and crack-free feature. 

A SAMs-based glucose biosensor was fabricated in stepwise process where two layers were casted on the surface of gold electrode (AuE). The main problem during electrochemical studies is peeling-off of enzyme film located at the electrode surface due to weak non-covalent interactions between GOx and Au surface. Therefore, strong bonding or protective layer is required to form SAMs. When the CS at a pH above its pKa value (6.3) it will deprotonate the primary amino groups and become insoluble with retention of its natural properties i.e., film forming ability and biocompatibility [144,158]. The development and characterization of SAMs at Au electrode for glucose biosensor was reported by Zhang et al., 2014 [158]. In this study, the working electrode was chemically modified using multilayered membrane structure with first layer of cysteamine (Cys) followed by GOx layer and CS as a final protective and adhesive layer. This sensor can be stored for up to 30 days, which may be attributed to the biocompatibility and strong interaction between AuE-Cys, Cys-GOx and CS-GOx. The hydrophilic CS can interact with GOx via –OH and –NH_3_^+^ side groups to form self-assembly for direct electron wiring to enhance the DET, as reported by Kumar-Krishnan et al. [101,102]. The enzymatic probe fabricated with CS stabilized bimetallic nanostructures with tailored geometries (Pd@Pt Core–Shell Nanocubes) was also reported by the same research group [101]. Silver nanowires (AgNWs) with high bio-affinity were stabilized in CS, which can covalently attach on GCE tip to improve the charge transportation and sensitivity for electrochemical detection of glucose [102]. Thus, the multi-layered configuration showed possibilities of the patterning of proteins and nanomaterials in CS-matrix. Furthermore, fast response time for enzymatic reaction with satisfactory sensor performance was noticed due to integrated CS with superior biological properties. Qui et al., demonstrated that the amino-functionalized ferrocene-conjugate Fe_3_O_4_@SiO_2_ nanoparticles (Fc-AFSNPs) could be effectively employed for the fabrication of reagentless glucose sensors by forming CS-GOx-enzyme nanocomposite films on magnetic carbon paste electrodes (MCPEs) [145]. For this purpose, CS-Fc-AFSNPs solution was spread evenly onto the well-polished, clean MCPE surface and allowed to form a FMC-AFSNPs/CS composite film. The GOx incorporated composite film was also casted on previous dried film. The entrapped enzyme and mediator showed efficient electronic communication and promising approach for construction of biosensor and bioelectronic devices for many analytes. In another study, GCE was modified in two steps, the electrochemical deposition of CS-CuNPs followed by cross-linking of GOx on CS-CuNPs-GCE surface using GA and bovine serum albumin (BSA). The resulting interface (CS-CuNPs/GOx-GA-BSA/SWCNTs-GCE) not only amplified the reduction current of H_2_O_2_ but also inhibited the responses of interferents at a much lower applied potential [158]. The synergistic electrocatalytic effect of CuNPs and SWCNTs exhibited good characteristics including a large determination range, good sensitivity, fast response time (<4 s), high stability, and excellent selectivity. Although SAMs provide many advantages, the blocking behavior of the SAMs may decrease electron-transfer rates significantly.

##### Layer by Layer

The alternate adsorption of polyanions and polycations with integration of different materials such as nanomaterials, proteins/ enzymes in matrix onto solid substrates in the form of thin film is so called the layer-by-layer (LBL). LBL is a self-assembly-driven surface modification strategy which allows the construction of multilayered nanostructured films onto substrates of any geometry, from simple bi-dimensional surfaces to more complex 3-D porous scaffolds [160]. Multi-layered structure can be formed by repeating the same procedure for desired configuration and morphological features. The layered architecture also as an additive and protective sheet which acts as a shield for protein conformation and prevent destabilization of active sites of the biomolecules. Super conductivity of the enzyme-wired CS-matrix is still big challenge for researchers to build-up fast and real time glucose biosensor. Multilayers structures can be produced using LBL technique which exponentially increased the chances of enzyme-binding to the matrix and its reusability, long term stability with regeneration and reusability of the transducing component. LBL with multilayer composition not only results in excellent electrochemical properties but also favors functionality of the designed assemblies. Chitosan′s inter-chain structure and its associations with other reactive groups in LBL can trigger a 3-D hydrogel network and accommodate/trap more enzymes easily [44,59,60,92,161]. An enzymatic glucose biosensor based on the self-assembling supramolecular LBL architecture comprising of CS-derivatives, sodium salt of 3-mercapto-1-propansulfonic acid, salt of 3-mercapto-1-propansulfonic acid, NF and GOx onto thiolated gold electrodes was reported by Miscoria et al., 2006 [92]. The bioelectrode modified with five quaternized CS/GOx bilayers exhibited highly selective response with zero percent interface effect from AA and UA. It has very good analytical performance with consumption of small amounts of reagents. Further, the credit also goes to the rational design of biorecognition layers where an alternate electrostatic adsorption of polyelectrolytes allows noticeable improvements in the selectivity and sensitivity of a biosensor with easy immobilization. Metallic nanostructured layer of copper oxide (Nano-CuO) sputtered thin film on the conductive fluorinated-tin oxide (FTO) layer that was exploited for covalent linkage of GOx via CS for impedimetric glucose biosensing. One –CHO group of glutaraldehyde linked covalently to –NH_2_ group of chitosan while other –CHO group is covalently linked to –NH_2_ groups on the surface of GOx. This mulit-layered electrode modification afforded excellent microenvironment for rapid biocatalytic reaction to glucose in real sample [139].

##### Sandwich Configuration

Biosensor with sol-gel matrix mostly suffer from the limited diffusion, slow response time and restricted electron transfer. The sandwich configuration can be better option to overcome these limitations where enzyme layer is protected by top and bottom layer of composites or nanocomposites. The typical sandwich configuration at electrode allows homogenous distribution of enzyme and nanomaterials with controlled matrix porosity. High amount of active enzyme triggers fast enzymatic catalysis and layered porous matrix provides rapid diffusion of substrates and analytes required [162,163]. Miao et al., 2000, reported the enzymatic sensor based on sandwich configuration using inner layer of CS-Fc and GA cross-linked GOx-CS as outer layer on CPE [100]. This sandwich biosensor proved that this kind of biosensors collaborate well with a classical UV spectrophotometric technique with a relatively fast response.

In some cases, a separate film/ membrane containing enzyme was placed on the surface of a moist dialysis membrane as a lamination layer and fastened with an O-ring [164]. While in most of the cases, CS layer on the electrode surface has the same function without the need of any extra binder. For example: carbon nanochips (CNCs) and GOx were fixed tightly using CS as binder to the surface of the GCE surface. The synergistic effect of CS and CNCs-GOx enhanced and promoted electron transfer thereby decreasing resistance [94]. This sandwich configuration also favors adsorption of GOx and retains its native secondary structure with excellent electrocatalytic properties.

#### 6.2.3. Microelectrode Arrays/Printing

Conventional electrodes suffer from sensitivity and sluggish electron transportation. Electrode micro-fabrication technology offers great way to overcome these issues. Such microelectrodes (µ-electrode) deal with issues like diffusion controlled current, low charging current, high signal to background ratio (Faradaic/charging), and reduced solution-resistance due to their highly compact-arrayed design with bulk production of multi-electrodes [165,166]. Multianalyte-sensing systems with improved durability and interference-free features can be developed using miniaturized electrodes and CS-matrix. Protective feature of the CS and its pH-dependent solubility were also confirmed by Voltage-dependent study of amine-rich CS-polysaccharide. The study revealed that once the CS has deposited on electrode surfaces, it can act as a potential interface between biological systems and microelectronic devices for wide range of applications [167,168,169]. Micro-electrode array-based glucose biosensors integrated with chitosan-modified interfaces are summarized in Table 3. Thin film printed electrodes produced by photolithography method were further modified by sol-gel solution of CS and enzyme for glucose sensing. Huang et al., 2013 reported an enzymatic µ-electrode array sensor prepared by photolithography. The interface surface was coated with a layer of GOx, entrapped in a three-dimensional network composed of CS and tetraethyl orthosilicate (TEOS) sol-gel [165]. The GOx-CS-TEOS interdigitated µ-electrode showed good sensitivity, selectivity and stability when optimized for maximum GOx loading, the applied voltages, the concentration of mediator and the pH for glucose sensing. The resulted stable and reproducible biosensor exhibited a good response to glucose with a wide linear range and the small Michaelis–Menten constant value. This enzymatic-array was tested for 100 cycles using CV and confirmed stability of the probe, and flexibility with high reproducibility. The as-prepared gold interdigital array also possessed superb resistance to oxidation and has great potential for applications in portable and disposable sensors [165]. GOx-embedded cylindrical carbon fiber-based µ-electrodes were fabricated through the reduction of BQ and CAM to produce uniformly-coated films with the highest throughput [161]. The long-term stability of such µ-electrodes was possibly due to strong chemical bonding between BQ-CS and GOx through free amines of lysine residues on the enzyme to the CS-matrix. Smith et al., 2018; demonstrated and confirmed that GOx immobilization by entrapment in a CS-hydrogel is another effective method for µ-electrode fabrication, which can be used for the measurements of brain glucose. Researchers combined GOx-modified carbon-fiber µ-electrodes with fast-scan voltammetry (CV) for real-time measurements of glucose in brain tissues [95]. The detection range of the sensor was from 0.2 to 50 mM glucose in vitro using the flow-injection apparatus. The experimental finding demonstrated that sensor prepared with GOx entrapped in CS-hydrogel was most effective for real-time monitoring of glucose with high sensitivity, stability, low cost, and nontoxicity. While in another case, aligned carbon nanotubes (ACNTs) modified electrode has been developed [120]. The ACNTs electrode for reagentless biosensors were developed by electrodeposition and encapsulation for direct electron transfer in redox reaction of FAD/FADH_2_. Artigues et al., proved that sensor architecture utilizing titanium dioxide nanotube arrays (TiO_2_NTAs) not only has features like simplicity and low cost but also can be used for measuring glucose in four different food products, including soft drinks, soy sauces, dairy products and tomato sauces [170].

This nanotube array-based amperometric glucose sensor was constructed by physical immobilization of GOx in CS hydrogel onto highly ordered TiO_2_NTAs. GOx with CS-Neodymium orthophosphate nanoparticles (NdPO_4_NPs) composite with high specific surface area offered sensitive, selective and stable enzymatic biosensor for determination of glucose in human plasma. The CV and EIS studies revealed that the modified electrode can also be used to immobilize other redox enzymes due to composite configuration [171]. The analytical parameters of the array-device were ensured that the new methodology was accurate and specific. The detection quality was reproducible and robust over the specified range of food samples for glucose.

### 6.3. Nano-Chitinous Material for Glucose Sensors

Like CS and CT precursors, their micro and nano-form are also promoting the compatibility with natural surroundings to the biological counterpart during conjugation [26,27,157,172]. The unique physicochemical and biological properties of these chitinous structures are improved when their size is reduced exponentially [86]. Due to the superior physicochemical, optical, catalytic and reactive properties, nanomaterials of CS and CT have a wide range of applications in areas such as pharmaceuticals, biomedical research, cosmetics, purification technology, and sensing devices [173,174]. These cost-effective small-size particles of CT/CS are outstanding stabilizing agents with film-forming capabilities and high absorption power. Due to their mechanical strength, nanofibers (NF) are used to form blended nanomaterials with synthetic polymers (poly(ethylene oxide) (PEO) [175], PANI [28,89,176], poly(vinyl alcohol) (PVA) [9,41,42,123,135,174], poly(L-lactide) (PLA), poly(glycolic acid) (PGA), polypyrrole (PPy) [124], polyvinyl pyrrolidone (PVP) [177], natural polymers (silk, cellulose, collagen, alginate, zein and agarose), and mineral (hydroxyapatite) [173,174,175,176,177,178]. The biocompatibility, non-toxicity and large surface area make these composite materials a potential scaffold for enzyme immobilization. For example: CS-NFs-AuNPs for cholesterol oxidase [179]; magnetic chitin nanofiber (CT-NF) composite for chymotrypsin [180] and CS-cellulose acetate for physical adsorption of protease [181], etc.

#### 6.3.1. Nanochitin

Dilute acid hydrolysis of chitin at high temperature resulted into nano-whiskers (CTNWs) with slender parallelepiped rods [182,183]. These CTNWs have been successfully explored in nanotechnology and biosensing application. Nakorn investigated the possibilities of CTNWs (size 300 nm) and chitosan nanoparticles (CSNPs, size 39 nm) as immobilization material for GOx under different conditions to fabricate an efficient biosensor with improved features. The CSNPs appeared to be a better support than CTNWs for GOx immobilization due to longer storage time and more GOx binding [172]. Non-enzyme glucose electrochemical sensor was reported by Solairaj et al., 2017 using copper nanoparticle immobilized CT-nanocomposite (CTNC-CuNP). This study also confirmed that CTNC-CuNPs could be a potential antimicrobial, nontoxic and low cost material to develop sensors [184]. The biodegradable, electro-active CT-NF films were explored for flexible piezoelectric transducers, and their ferroelectric characteristic was confirmed by polarization measurements [183].

#### 6.3.2. Nanochitosan

Nanochitosan has very low toxicity and excellent adhesion property for enzyme immobilization and interface design for various applications [184,185,186,187,188,189,190]. Easy and convenient methodologies for CS-micro-particle preparation such as coagulation/precipitation, covalent cross-linking, ionic cross-linking, ionic gelation, polyelectrolyte complexation, coacervation or phase separation and emulsion droplet coalescence have been used for the synthesis of nanochitosan [187]. In one case, the tree-like structure of core–shell hyper-branched chitosan nanoparticles were prepared using nanospray dryer. The electrocatalytic performance of the developed bio-probe for fast electron communication between the enzyme′s active sites and the screen printed electrode was attributed to high surface area provided by branched CS particles. Such new architectures carrying a huge number of positive charges around the shell by creating more amino and hydroxyl groups can be more useful for enzyme functionalization [122]. The CS nano-layers was prepared using LBL thin film technique and GOx was adsorbed on these layers for the detection of glucose in body fluids, beverages and foodstuffs [162,170]. The relationship between frequency variation and number of bilayers of CS-GOx nanolayers deposited on platinum surface was studied and its morphological features were observed under atomic force microscope. The adsorption of positively charged CS on Pt electrode facilitated enzyme immobilization due to GOx’s negative charge, and CS also provided a favorable microenvironment for GOx to oxidize glucose and exchange electrons with underlying electrodes.

## 7. Future Prospectus

The reactive functional groups and surface charge of chitin have been successfully utilized for preparation of a wide spectrum of its derivatives such as chitosan, alkyl chitin, sulfated chitin, dibutyryl chitin carboxymethyl chitin, and nanostructured chitin/chitosan with high commercial values. However, there are limited studies based on chitin nanocomposites for biosensing application due to its insolubility in organic and inorganic solvent, low metal ion capacity, separation difficulty, non-tunable porous structure and low surface area. Still there is strong need for the development of simple and easy synthesis methodology for chitin composites and nanocomposites that can be used for applications for industrially important enzyme immobilization and practically useful biosensor fabrication. Ecofriendly and renewable chitinous materials when redesigned with nanomaterials, have the potential to become new, superior and advance materials. These nanocomposites could possess important properties such as compatibility, biodegradability, and mechanical strength for biosensors. In the future, these innovative interfaces with further tuning can be used for the modification of different kinds of transducers for versatile sensing with relatively short analysis time.

## Figures and Tables

**Figure 1 polymers-11-01958-f001:**
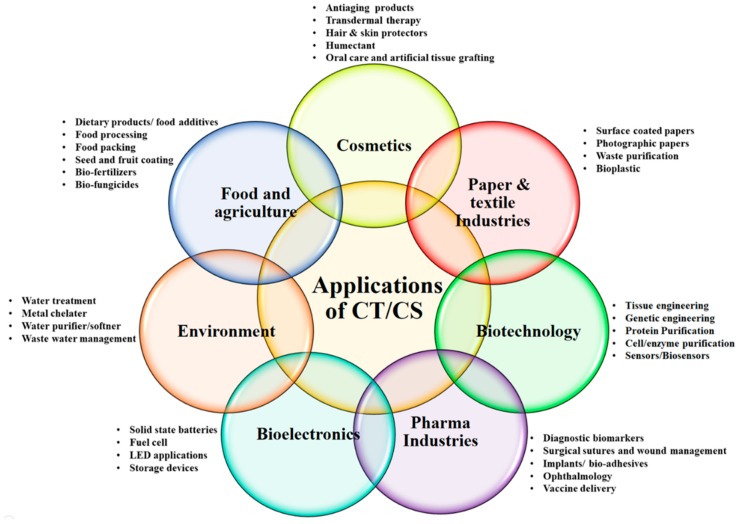
Multi-functionalities of chitosan (CS), chitin (CT) and their derivatives.

**Figure 2 polymers-11-01958-f002:**
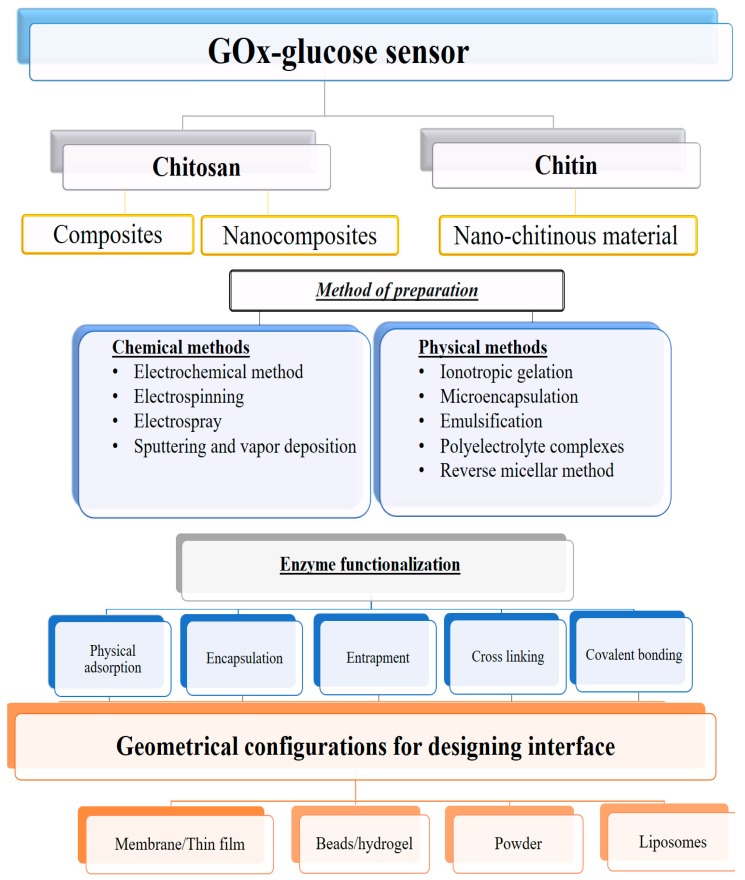
An overview of chitin and chitosan-based glucose biosensors using glucose oxidase.

**Figure 3 polymers-11-01958-f003:**
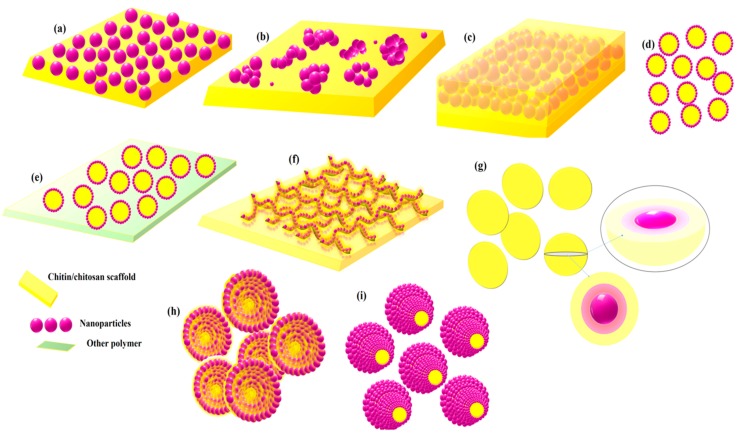
Role of chitinous matrix in the stabilization of nanostructures: (**a**) uniform stabilization, (**b**) non-uniform stabilization, (**c**) fully merged nanoparticles, (**d**) CS-coated nanomaterial free vesicles, (**e**) CS-coated nanomaterial vesicles stabilized in another polymer/CS-blend, (**f**) Nanoparticle decorated nanotubes network dispersed in CS-matrix, (**g**) encapsulated CS-nanomaterials tablets, (**h**) micellar structures with CS-coating and (**i**) CS-filled micellar structures.

**Figure 4 polymers-11-01958-f004:**
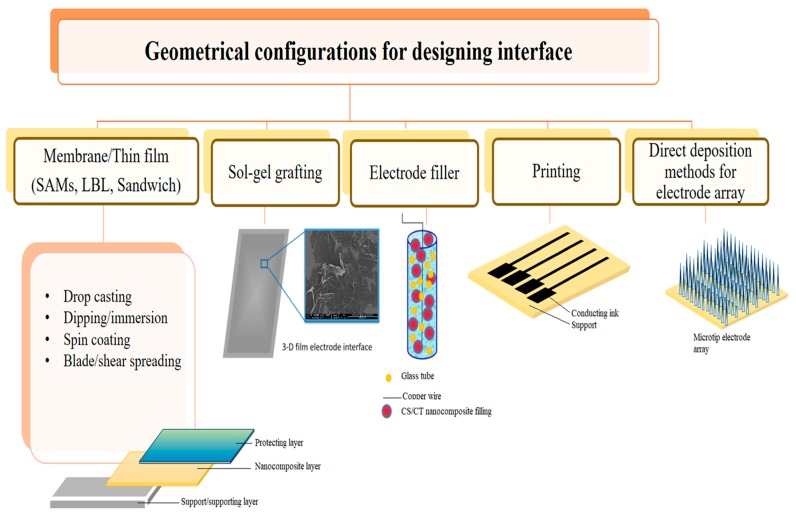
Illustration of the interface configuration in various format using chitin and chitosan for electrochemical glucose biosensing.

**Figure 5 polymers-11-01958-f005:**
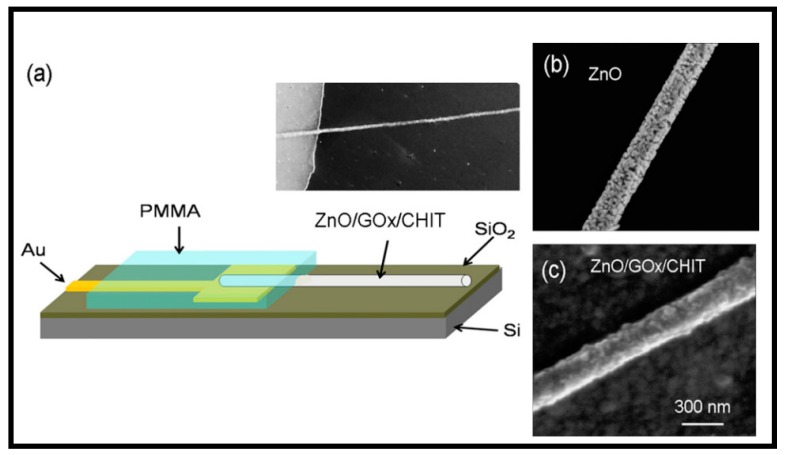
Construction of GOx sensor: single ZnO nanofiber grown on SiO_2_ layer and Au lead (**a**); SEM image of nanofiber without (**b**) and with GOx (**c**). Reproduced with permission from Ref. [116].

**Figure 6 polymers-11-01958-f006:**
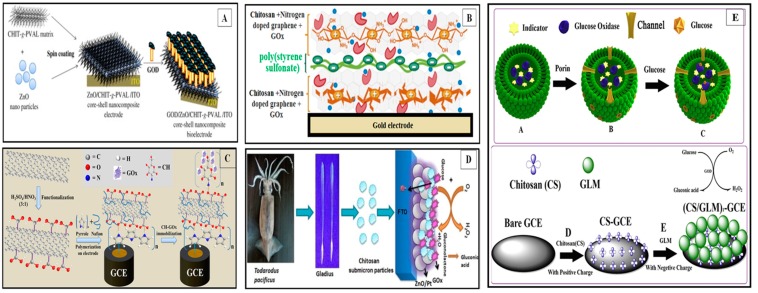
Some of the examples of different strategies reported for the modification of electrode surface: (**A**) spin-casted biocompatible core-shell porous CS-nanocomposite modified zinc oxide/platinum electrode [86]; (**B**) highly stable self-assembled layer by layer formation of N-doped enzyme matrix on indium-tin oxide glass substrate [123]; (**C**) encapsulated bio-nanohybrid film formation at GCE [140] (**D**) CS-submicron particles with GOx for amperometric glucose biosensor [86] and (**E**) modification of GCE using CS-GOX-liposome microreactors nanocomposite for electrochemical biosensor [146]. Reproduced with permission [86,123,140,144,146].

**Table 1 polymers-11-01958-t001:** Chitosan/chitin composite-based glucose oxidase (GOx) sensors for glucose detection.

Immobilization Matrix Composition		Method of Preparation	Sensing System	Characteristic Features/Application	Reference
Primary Phase	Secondary Phase				
Chitin	GOx	Adsorption based on electrostatic interactions	CS-GOx/CPE	Glucose detection in sports drink	[59]
Chitin	GOx	Adsorption based on electrostatic interactions	CS-GOx/PtE	Glucose detection in sports drink	[60]
Chitosan	GDI-AY9-GOx	Cross-linking	CS-GDI-GOx-AY9/PtE	New composite composition for CS-film, simple, efficient, and cost-effective enzyme immobilization, Standard glucose detection with linear range = 10 µM–5.0 mM and LOD = 10 µM	[81]
Chitosan	CS-PNMP-GOx	Cross-linking	CS-PNMP-GOx/PtE	Standard glucose detection	[82]
Chitosan	TOES	Sol-gel encapsulation	CS-GOx-TOES/GCE	Standard glucose detection	[84]
Chitosan	SiO_2_-GOx	Sol-gel entrapment	CS-SiO_2_-GOx/PB-NF/GCE	Glucose detection in human blood samples	[85]
Chitosan & pH sensitive polymer	GOx-CAT	Cross-linking		Urine glucose detection	[88]
Chitosan	GOx	Absorption	CT-GOx/PtE	Layer-by-layer thin films, Standard glucose detection	[91]
Chitosan	Thiolated gold-GOx	Adsorption	CS-GOx-MPS/ CHIT/Naf/AuE	Human Serum glucose detection	[92]
Chitosan	GOx-DNA	Adsorption	CS-GOx-DNA/GCE	Standard glucose detection	[93]
Chitosan	GOx-CNCs	Adsorption	CS-GOx/CNCs/GCE	Standard glucose detection	[94]
Chitosan	GOx	Adsorption Hydrogel entrapment Nanofibers entrapment	CS-GOx CS-GOx CS-PVA-GOx	Brain glucose detection	[95]
Chitosan	Cos-GOx	Physical mixing	Cos-GOx-Ferri/SPCE	Standard glucose detection	[96]
Chitosan	Pb-G-GOx	Sol-gel adsorption	CS-GOx/PB-G/PS-StE	string sensor with PB modified graphite and CS, linear range = 0.03 to 1.0 mM, LOD =10 µM Glucose detection in spiked human serum samples	[97]
Chitosan	PB-GOx PB-GalOD PB-GluOD	Cross-linking	CS-GOx-PB/PtE CS-GalOD-PB/PtE CS-GluOD-PB/PtE	Human blood serum and fermented solution Glucose, galactose glutamate detection	[98]
Chitosan	GOx	Cross-linking	CT-GOx/PtE	Amperometric biosensor Glucose detection in beverage drink samples	[99]
Chitosan	Fc-GOx	Sandwich configuration with cross-linking	CS-Fc:GA-GOx-CS/CPE	Fast response time, Linear range = 8 × 10^−4^ to 1.7 × 10^−2^ M, LOD = 8 × 10^−4^ M, Glucose in soft drink samples	[100]
Chitosan	AgNWs-GOx	Covalent linkage	CS-AgNWs-GOx/GCE	Standard glucose detection	[101]
Chitosan	Pd@PtNC-GOx	Covalent immobilization	CS-GOx/Pd@Pt NC/GCE	Standard glucose detection	[102]
Chitosan	HRP-GOx	Electrodeposition and Covalent coupling sol-gel	CS-GPTMS-GOx- HRP/AuE	Standard glucose detection	[103]

Note: Acetyl Yellow 9-AY9, Catalase—CAT, Calcium alginate—CA, Carbon nanochips (CNCs), Carbon paste electrode—CPE, Carbon fiber—CF, Cos—chitosan oligomers, Ferrocene—FC, Galactose oxidase—GalOD, Grafted poly (vinyl alcohol)—gPVA, Glutamate oxidase (GluOD), γ-glycidoxypropyltrimethoxysiloxane—GPTMS, Glutaric dialdehyde—GDI, polyester spun—PS, Prussian blue—PB, poly(N-methypyrrol)—PNMP, Platinum electrode—PtE, Sodium salt 3-mercapto-1-propansulfonic acid—MPS, String electrode—StE, Traethylorthosilicate—TEOS.

**Table 2 polymers-11-01958-t002:** Chitosan/chitin nanocomposite-based glucose biosensors.

Conjugation Method	Chitinous Sensing System	Reinforced Secondary Phase	Linear Dynamic Range	LOD	Target Sample	Reference
Electrostatic adsorption	CS-GOx/AuNPs/PAA/PtE	AuNPs-GOx	0.5–16 mM	7.0 µM	Human serum glucose	[44]
Encapsulation	CS-κ-Cg-GOx/AuNPs/AuE CS-κ-Cg-GOx/AuE	CNT-PtNP-MTOS	10 µM–7.0 mM 10 µM–7.0 mM	5.0 µM 5.0 µM	Spiked saliva glucose	[70]
Adsorption	GCSPs-GOx-(ZnO-Pt) NPs/FTOE SCS-GOx-(ZnO-Pt)NPs/FTOE	GCSP-GOx	0.05–1.0 mM 0.05–1.0 mM	0.22 mM 0.31 mM	Standard glucose	[86]
Cross-linking	CS-GOx /PtNPs/SCS/ZnO CS-GOx/PtNPs/GCSP/ZnO	PtNPs-GOx	0.05–1.0 mM 0.05–1.0 mM	0.09 mM 0.053 mM	Standard glucose	[87]
Absorption	PANI-SnO_2-_NF/GOx-HRP-CS/GCE	GOx-HRP-CS	5.0–100 μM	1.8 μM	Spiked human urine glucose	[89]
Adsorption	CS-GOx-DNA/GCE	GOx-DNA	0.04–2.28 mmol L^−1^	0.04 mmol·L^−1^	Standard glucose	[93]
Covalent bonding	GOx-CDI/CS-CNTs-GA/PANI-AuE	GOx-CS-CNTs	1.0–20 mM	1.0 mM	Standard glucose	[105]
Covalent linkage	CS-G-MNPs-GOx/Pt-ITOE	MNPs-GOx	16 μM–26 mM	16 μM	Standard glucose	[107]
Entrapment & cross linking	CS-GOx-SWNTs/E	GOx-SWNTs	10 µM–35 mM	2.5 µM	Standard glucose	[112]
Covalent linking	CS-CNT-GOx-Fc-RD/E	GNPs-GOx	0.02–2.91 mM	7.5 μM	Human blood glucose	[113]
Entrapment	CS-GR70-GOx-NF/GCE	CS-GR70-GOx	0.14–7.0 mM	17.5 mM	Standard glucose	[114]
Adsorption	CS-G-GOx/GCE	G-GOx	0.08–12 mM	0.02 mM	Standard glucose	[115]
Electrostatic adsorption	CS-ZnONF-GOx/E	ZnONF-GOx	0.2–12 mM	0.2 mM	Intra cellular glucose	[116]
Electrochemical deposition	CS-GOx/Fe_3_O_4_NPs-AuNPs/AuE	Fe_3_O_4_Nps-AuNPS-GOx	3.0 μM–0.57 mM	1.2 μM	Human blood glucose	[117]
Electrostatic interactions	CS-rGO-Con A/GCE	Con A-rGO	1.0−10.0 mM	1.0 mM	Glucose, Urea	[118]
Electrochemical deposition	CS-AuNPs-GOx/GTE	AuNPs-GOx	0.616–14.0 mM	0.202 mM	Blood serum glucose	[119]
Encapsulation	CS-gPVA-ZnONPs/GOx/ITOE	gPVA-ZnONPs-GOx	2.0 μM–1.2 mM	0.2 µM	Blood serum, urine glucose	[123]
Entrapment	CS-PPyNTs-AuNPs-GOx/ITOE	PPyNTs-AuNPs-GOx	3.0–230 μM	3.10 μM	Standard glucose	[124]
Adsorption	CS-GOx-rGO_(HHA)_-ZnO-AgNPs/GCE	GOx-rGO_(HHA)_-ZnO-AgNPs	0.1–12 mM	10.6 μM	Blood serum glucose	[125]
Electrostatic adsorption	CS-Fc-GONS-GOx/GCE	Fc-GONS-GOx	0.02–6.78 mM	7.6 μM	Standard glucose	[126]
Adsorption	CS-FGS-PtNPs/CS-GOx/GCE	FGS-PtNPs-GOx	0.3 μM–5.0 mM	0.6 μM	Blood glucose	[127]
Electrostatic adsorption	CS-GOx-PtNPs/ZnO-FTOE CS-GOx/ZnO-FTOE	CS-GOx-PtNPs CS-GOx	16.6 µM–2.0 mM 31.19 µM–2.0 mM	16.60 µM 31.19 µM	Standard glucose	[128]
Entrapment	CS-SiO_2_-GOx-Nf-Pt/MWNTs/GCE	SiO_2_-GOx	1.0 μM–23 mM	1.0 μM	Standard glucose	[129]
Entrapment	CS-GOx/TEOS-APTES-Fc-GONS/GCE	GOx/TEOS-APTES-Fc-GONS	0.02–5.39 mM	6.5 mM	Blood serum glucose	[130]
Adsorption	CS-G-AuNPs-GOx/AuE	G-AuNPs-GOx	2.0–14 mM	180 μM	Blood glucose	[131]
Electrochemical deposition	CS-GOx-(Au-PB) NPs/GCE (Au-PB)NPs-GOx/GCE	GOx-(Au-PB)NPs	0.2–3.0 × 10^−3^ M 0.2–1.9 × 10^−3^ M	0.2 mM 0.2 mM	Standard glucose	[132]
Electrochemical deposition	CS-Fc/AuNPs/GOx/GCE	Fc-GOx	0.02–8.66 mM	5.6 µM	Serum glucose	[133]
Cross-linking	CS-Fe_3_O_4_-AuNPs-GOx/GrE	Fe_3_O_4_-AuNPs-GOx	5.0–30 mM	0.55 mM	Blood glucose	[134]
Adsorption	CS-GOx-Fe_3_O_4_NPs/ Au-coated glass E	GOx-Fe_3_O_4_NPs	1.0 × 10^−6^–3.0 × 10^−2^ M	0.04 mmol·L^−1^	Standard glucose	[135]
Electrodeposition	CS-g-PAN-GOx/PtE	g-PAN-GOx	0.5–16 mM	0.5 mM	Standard glucose	[136]
Adsorption	CS-GOx/Fe_3_O_4_/ITOE	Fe_3_O_4_NPs-GOx	10–400 mg dL^−1^	0.5 mM	Standard glucose	[137]
Adsorption	Cathode; CS-GQDs-AuNPs/PDDA-MWCNTs/CS/CBC and Anode: GOx-CBA	CS-GQDs-AuNPs-PDDA-MWCNTs	0.1–5000 μM	64 nM	Blood glucose	[138]
Cross linking	CS-GOx/Nano-CuO-FTOE	CS-GOx	0.2–15 mM	27 μM	Blood serum glucose	[139]
Electrostatic adsorption	CS/GOx/GNPs/Ppy-Nf-fMWCNTs/GCE	FMCNTs-GOx	5.0 µM–4.7 mM	5.0 µM	Human serum glucose	[140]
Cross-linking	CS-GOx/ZrO_2_/NF/PtE	ZrO_2_-GOx	1.25 × 10^−5^–9.5 × 10^−3^ M	1.0 × 10^−5^ M	Blood glucose	[141]
Entrapment	CS-GOx/ MnO_4_NPs/AuDE	GOx-MnO_4_NPs	NA	NA	Standard glucose	[142]
Entrapment	CS-NG-GOx-PSS/AuQC CS-GOx-PSS/AuQC	NG-GOx GOx-PSS	0.2–1.8 mM 0.2–1.8 mM	64 μM 112 μM	Standard glucose	[143]
Entrapment	CS-GOx-FMC-AFSNPs/MCPE	GOx-FMC-AFSNPs	1.0 × 10^−5^ –4.0 × 10^−3^ M	3.2 μM	Standard glucose	[144]
Encapsulation and entrapment	CS-GOx-LM/GCE	CS-GOx-LM	0.01–10 mmol·L^−1^	1.31 μmol·L^−1^	Food sample-Fruit juice glucose	[145]
Electrodeposition	CS-GOx/Au-PtNPs-CNTs/GCE	Au-PtNPs-CNTs-GOx	0.001–7.0 mM	0.2 μM	Human blood, urine	[146]
Electrochemical deposition	CS-GOx/AuNPs/GCE	AuNPs-GOx	5.0 × 10^−5^–1.30 × 10^−3^ M	13 μM	Standard glucose	[147]
Electrochemical deposition	CS-AuNPs-GOx/PB-GCE	AuNPs-GOx	1.0 × 10^−6^–1.6 × 10^−3^ M	6.9 × 10^−7^ M	Human serum glucose	[148]
Electrodeposition	CS-GOx/AuNPs/AuE	AuNPs-GOx	5.0 μM–2.4 mM	2.7 µM	Serum glucose	[149]
Cross-linking	GOx-(CS-ZnO)NS-NF/PtFe(III)	ZnONS-GOx	10 μM–11.0 mM	1.0 µM	Standard glucose	[150]
Electrodeposition	CS-GOx-MWCNTs/AuE	GOx-MWCNTs	5.0 µM–8.0 mM	6.8 mM	Standard glucose	[151]
Electrodeposition	CS-GOx-Pt–PbNPs/SSNE	GOx- Pt–PbNPs	0.03–9.0 mM	0.03 mM	Standard glucose	[152]
Electrodeposition	CS-GOx-IL-MWCNTs/nanoAuE	GOx-IL-MWCNTs	3.0 µM–9.0 mM	1.5µM	Serum glucose	[153]
Adsorption	CS-GOx-AgNWs/GCE	GOx-CS-AgNWs	10 μM–0.8 mM	2.83 µM	Spiked serum glucose	[154]
Encapsulation	CS-GOx/CNT-PtNP-MTOS/GCE	CNT-PtNP-MTOS	1.2 × 10^−6^–6.0 × 10^−3^ M	3.0×10^−7^ M	Human serum glucose	[155]
Encapsulation	1. CSNPs-GOx/AuE 2. CS-GOx/AuE	CSNPs-GOx	0.001–1.0 mM	1.1 mM	Standard glucose	[156]
Covalent bonding	CS-Cys-GOx/AuE	Cys-GOxc126	10.5–27 mM	316.8 μM	Standard glucose	[157]

Note: Chitin nanocomposite—CTNC, 1,4-carbonyldiimidazole—CDI, Ferricyanide—FCN, Ferrocene—Fc, Fluorine doped tinoxide electrode—FTOE, Functional graphene sheets—FGS, Gladius chitosan submicron particles—GCSPs, Gold disk electrode—AuDE, Gold–Platinum alloy nanoparticles—Au-PtNPs, Gold-Prussian blue nanoparticles—(Au-PB)NPs, Grafted dendrimer—RD, Graphene tape electrode—GTE, Graphite Rod—Gr, Hydrazine hydrate—HHA, IL-Ionic liquid Iron oxide nanoparticles—Fe_3_O_4_NPs, kappa-carrageenan—κ-Cg, Liposome microreactor—LM, Magnetic nanoparticles—MNP, Microelectrode—µE, Monocarboxylic acid—FMC, Nafion—Nf, Nanofiber—NF, Nanosheet—NS, Polyaniline—PANI, Poly(allylamine)—PAA, Poly(allylamine)—PAA, Porous graphene—GR, Standard chitosan (SCS), Stainless steel needle electrode—SSNE.

**Table 3 polymers-11-01958-t003:** Micro-electrode array-based glucose biosensors integrated with chitosan-modified interfaces.

Sensing System	Method of Preparation	Linear Dynamic Range	Sensitivity	LOD	Target Sample	Reference
(CS-PVA)-GOx	Nanofibers entrapment	0.2–50 mM	~0.4–15 nA·mM^−1^	~0.6–1.0 mM	Brain glucose	[95]
CS-GOx-CdS/ACNTs-Pt_nano_/GCE	Electrodeposition and encapsulation	400 μM–21.2 mM	1.0 µA·mM^−^^1^	46.8 μM	Standard glucose	[120]
CS-BQ-GOx/Au-µE CS-CFM0-GOx/ Au-µE	Covalent bonding	0–1.6 mM	14.4 nA·mM^−1^ 13.5 nA·mM^−1^	8.9 µM 11.5 µM	Standard glucose	[160]
(CT-GOx)_n = 6_/PtE	Absorption Layer-by-layer thin films	NA	NA	NA	Standard glucose detection	[162]
CS-TEOS-GOx/Au-SiO_2_µE	Entrapment-So-gel	0–35 mM	8.74 µA·mM^−^^1^·cm^2^	1.0 mM	Standard glucose	[165]
(CS-PVA-GO)Nf-GOx/PtE	Cross-linking & co-electrospinning	5.0 μM–3.5 mM	11.98 µA·cm^−1^·mM^−1^	5.0 μM	Human serum glucose	[169]
(CS-GOx)/TiO_2_NTAsE	Physical entrapment-hydrogel	0.3–1.5 mM	5.46 µA·mM^−^^1^	0.07 mM	Soft drinks, Dairy products, tomato & soy sauces	[170]

Note: Not available—NA, aligned carbon nanotubes—ACNTs, benzoquinone—BQ, Nafion—Nf, Polyvinyl alcohol—PVA, TEOS—Tetraethyl orthosilicate, Titanium dioxide nanotube arrays—TiO_2_NTAs.

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
