# Peer review of "Exploration of Chitinous Scaffold-Based Interfaces for Glucose Sensing Assemblies"

_polymers, 2019, doi:10.3390/polym11121958_

Round 1

Reviewer 1 Report

This manuscript by Bagal-Kestwal and Chiang reviewed on the chitinous scaffold-based interfaces for glucose sensing assemblies is straightforward and attractive.

Author Response

Reviewer 1: This manuscript by Bagal-Kestwal and Chiang reviewed on the chitinous scaffold-based interfaces for glucose sensing assemblies is straightforward and attractive.

Answer: Thank you very much! We deeply appreciate the review comments and suggestions by all reviewers which have helped to improve the quality of this review manuscript. All changes are highlighted in blue in the revised manuscript.

Thank you for your encouraging words.

Reviewer 2 Report

The manuscript fulfills the requirements to be published in Polymers.

However, some minor corrections should be made prior to acceptance of the manuscript.

In  line 565” called the layer-by-layer (LbL). LbL is a self-assembly-driven surface modification strategy which” and in line  573 “ using LBL technique”

LbL and LBL , are the same? Why do you use different notation?

In line 379 “ Nafion (NF), butyl carbitol acetate, dimethylformamide, cellulose..”

And in 670-671 “..670 nanofibers 671 (NF)”

It is confusing.  Instead of Naftion ( NF ) you could use (Nf)

In Figure captions is missing:

Table 3: Micro-electrode array based glucose biosensors integrated with chitosan-modified interfaces

Author Response

Response to Reviewer 2 Comments:

The manuscript fulfills the requirements to be published in Polymers. However, some minor corrections should be made prior to acceptance of the manuscript.

Q1. In line 565” called the layer-by-layer (LbL). LbL is a self-assembly-driven surface modification strategy which” and in line 573 “using LBL technique” LbL and LBL, are the same? Why do you use different notation?

Answer: We have replaced the LbL to “LBL” in the revised manuscript. Thank you!

Please see P. 39, Line 565.

 Q2. In line 379 “Nafion (NF), butyl carbitol acetate, dimethylformamide, cellulose..” And in 670-671 “..670 nanofibers 671 (NF)” It is confusing.  Instead of Naftion (NF) you could use (Nf).

Answer: As per your suggestion, we have changed the abbreviation for Nafion as (Nf) to avoid confusion with nanofiber (NF). Thank you for your suggestion.

Please see P. 27, Line 379. Table 2- P. 32-34 [Ref. 129 and 140].

Q3. In Figure captions is missing: Table 3: Micro-electrode array based glucose biosensors integrated with chitosan-modified interfaces

Answer: Thank you! We have listed the caption for Table 3 in “Figure caption” section. 

Please see P. 75, Line 305-306. Thank You!

Reviewer 3 Report

The review is well written and structured. However, the following points should be addressed before publication:

The concept of direct electron transfer (DET) with GOx should be revised. Within the text the possibility of DET with GOx is mentioned a couple of times. However, it is well known that as a result of the FAD sub-unit being deeply buried inside the protein shell, GOx is not capable of providing DET (see recent literature about it). Please amend the text as required.

Revise in detail the information provided in the tables. For example, in Table 1 some abbreviations are not clear (CNCs is not mentioned in the note, TOES is used in the table entry while TEOS is used in the note?). Moreover, different abbreviations are sometimes used for the same definition (for example: Cs, CHIT, CS). For all tables, please keep consistency for the abbreviations and use the same as used in the text. For each table, include all abbreviations used in the corresponding note at the end.

Figure 6 is not properly described. The figure consists of several panels that should be particularly mentioned in the text when they are described (Figure 6A, Figure 6B, ...). A better description of the different panels should also be provided in the figure caption, as well as an unequivocal indication to the references from where the individual figures where taken.

Please revise the use of abbreviations in the whole text: keep consistency and include a definition the first time that each abbreviation is used.

Author Response

Response to Reviewer 3 Comments:

The review is well written and structured. However, the following points should be addressed before publication:

Q1. The concept of direct electron transfer (DET) with GOx should be revised. Within the text the possibility of DET with GOx is mentioned a couple of times. However, it is well known that as a result of the FAD sub-unit being deeply buried inside the protein shell, GOx is not capable of providing DET (see recent literature about it). Please amend the text as required.

Answer: Thank you! We agree with the reviewer and have amended the information in “Innovative strategies for sensor interface” section. Please see P. 19, Line 263-265 and P. 21, Line 267-285.

Q2. Revise in detail the information provided in the tables. For example, in Table 1 some abbreviations are not clear (CNCs is not mentioned in the note, TOES is used in the table entry while TEOS is used in the note?). Moreover, different abbreviations are sometimes used for the same definition (for example: Cs, CHIT, CS). For all tables, please keep consistency for the abbreviations and use the same as used in the text. For each table, include all abbreviations used in the corresponding note at the end.

Answer: We have corrected the misused abbreviations and highlighted the corrections in the revised manuscript. All of the abbreviations in the revised manuscript should be consistent now. Thank you!

Q3. Figure 6 is not properly described. The figure consists of several panels that should be particularly mentioned in the text when they are described (Figure 6A, Figure 6B, ...). A better description of the different panels should also be provided in the figure caption, as well as an unequivocal indication to the references from where the individual figures were taken.

Answer: For better understanding, we have elaborated the caption for figure 6 and incorporated brief description for each panel in the revised manuscript. (P. 28-29, Line 405-410 and P. 76-77, Line 1301-1309). Thank you.

Q4. Please revise the use of abbreviations in the whole text: keep consistency and include a definition the first time that each abbreviation is used.

Answer: We have corrected and revised the abbreviations throughout the manuscript. The definitions of abbreviations are also provided when they are first used in the text. The corrections are highlighted in the revised manuscript. Thank you for your valuable suggestions.